# Influence of Cheese Composition on Aroma Content, Release, and Perception

**DOI:** 10.3390/molecules29143412

**Published:** 2024-07-20

**Authors:** Isabelle Andriot, Chantal Septier, Caroline Peltier, Elodie Noirot, Pascal Barbet, Romain Palme, Céline Arnould, Solange Buchin, Christian Salles

**Affiliations:** 1Centre des Sciences du Goût et de l’Alimentation, CNRS, INRAE, Institut Agro, Université de Bourgogne, F-21000 Dijon, France; 2CNRS, INRAE, PROBE Research Infrastructure, ChemoSens Facility, F-21000 Dijon, France; 3Plateform DimaCell, Agroécologie, INRAE, Institut Agro, Université Bourgogne Franche Comté, F-21000 Dijon, France; 4Procédés Alimentaires et Microbiologiques, INRAE, Institut Agro, Université de Bourgogne, F-39800 Poligny, France

**Keywords:** cheese, aroma compounds, sensory analysis, PTR-MS, TCATA, GC–MS, image analysis

## Abstract

The quality of a cheese is determined by the balance of aroma compounds primarily produced by microorganisms during the transformation of milk into ripened cheese. The microorganisms, along with the technological parameters used in cheese production, influence aroma formation. The perception of these compounds is further influenced by the composition and structure of the cheese. This study aimed to characterize how cheese composition affects aroma compound production, release, and perception. Sixteen cheeses were produced under controlled conditions, followed by a quantitative descriptive analysis post ripening. Aroma composition was analyzed using HS-SPME–GC–MS, and a dynamic sensory evaluation (TCATA) was combined with nosespace analysis using PTR-ToF-MS. Image analysis was also conducted to characterize cheese structure. Cheese fat and whey lactose contents were identified as key factors in the variability of sensory attributes. GC–MS analyses identified 27 compounds correlated with sensory attributes. In terms of aroma compound release, 23 ions were monitored, with fat, salt, and lactose levels significantly affecting the release of most compounds. Therefore, cheese fat, salt, and whey lactose levels, as well as the types of microbial strains, play a role in influencing the composition, structure, release of aroma compounds, and sensory perception.

## 1. Introduction

The quality of a cheese aroma (retronasal odor perception) is determined by the balance of volatile compounds produced mainly by microorganisms during the transformation of milk into ripened cheese. Inoculations of various lactic bacteria that make up the secondary microflora, which produce flavor compounds [1,2,3,4,5], could be a technological lever to modify the composition of such compounds in cheeses. For example, Sgardi et al. [6] have demonstrated various aromatic potentials in model media. Aromatic potentials may be expressed in cheese as a function of the technological pathway [7]. Understanding and controlling this is currently an important variable in order to modulate the production of flavor compounds. The release of aroma compounds depends on the composition, structure, and texture of the matrix [8,9,10,11]. The perception of this complex aroma may depend on the mixture of volatile compounds present in the matrix, but their release in the mouth also depends on judges’ physiological factors linked to food breakdown such as mastication, swallowing, oral volume, velum opening, and salivary flow and composition under the chewing effect [12,13,14,15,16,17,18,19,20,21].

Many studies have shown the importance of the consumers’ oral physiology, as well as the composition and texture of the food matrix, on the release and temporal perception of flavor compounds [8,17,22,23]. The phenomena leading to the release of volatile compounds in the mouth are complex [24]. The real impact of each parameter, including composition and oral physiology, and their interactions on the release kinetics of each compound in the mouth, are still poorly understood, and some of the observed effects are difficult to interpret [25].

Thus, the relationships between the properties of the food matrix and the phenomena of stimulus release and perception are not yet well understood. Furthermore, in most studies, the matrices studied are model lipoprotein gels, whose structures differ greatly from those of real cheeses. Additionally, the results of the flavor compounds’ release into a solid matrix, obtained from compounds initially added to milk, may not be extrapolated to those synthesized in situ by microorganisms. For instance, Repoux et al. [26] studied the influence of the properties (firmness and fat content) of a solid processed model cheese on in vivo aroma release, considering the role of the in-mouth process during both mastication and post-swallowing steps, and the hydrophobicity of two added aroma compounds. Ethyl propanoate showed a higher release rate for firmer cheese and was more abundantly released during the mastication step, whereas nonan-2-one was more abundantly released during the post-swallowing step and remained more persistent in the mouth due to its higher hydrophobicity. To the best of our knowledge, no studies have been carried out on real cheeses to explain the relationships between the release of aroma compounds and salt, and their perception.

The objective of this study was to characterize the influence of fat, whey lactose and salt levels, and the type of strains on aroma formation of cheeses by HS-SPME–GC–MS, and through a sensory description of the cheeses. The release of the cheese aroma compounds and the perception of these were also analyzed using the physicochemical technique (PTR-ToF-MS, proton transfer reaction-time of flight-mass spectrometry) coupled with a temporal sensory analysis (TCATA, temporal check-all-that-apply) [27,28].

## 2. Results

### 2.1. Gross Composition of Cheeses

The cheese-making process was designed to modify the cheese composition through a two-level formulation of four basic technological parameters. The two levels of the cheese fat/dry matter ratios achieved were 41.0 ± 0.5 and 50.3 ± 1.0% (targets of 40% and 50%). Additionally, dry matter and all calculated parameters, including fat, were affected. The modification of whey lactose content (33 g·L^−1^ and 42 g·L^−1^) resulted in the expected difference in cheese pH upon unmolding (5.23 ± 0.08 and 5.16 ± 0.08, respectively) and at the end of ripening (5.38 ± 0.07 and 5.22 ± 0.09). The ratios of NaCl/water in the cheese were 2.6 ± 0.4 and 4.2 ± 0.4% (for 2.5 and 4.0%). They were linked to differences in proteolysis (NPN/TN of 11.78 ± 0.61% and 10.91 ± 0.66%, respectively) and mineralization (Ca/NonFat Matter of 4.54 ± 0.20 and 4.39 ± 0.17%). Lastly, as expected, the addition of the strains A1 and A2 did not affect any gross composition parameters except pH upon unmolding (5.14 ± 0.06 and 5.25 ± 0.05). All these differences were found significant through a four-factor analysis of variance (not shown, *p* value < 0.05). They are represented in Figure 1, based on the chemical composition (in red). Axis 1 represents the Fat/DM factor (G1 vs. G2), axis 2 represents the salt/water factor (S1 vs. S2) and axis 3 the lactose content of whey (T1 vs. T2).

### 2.2. Taste and Texture of Cheeses

The quantitative descriptive analysis (QDA) provided a sensory description of the cheeses using 30 aroma descriptors, 7 taste descriptors, 1 intensity descriptor and 6 texture descriptors. The 16 cheeses were described differently, depending on their composition. Figure 1 shows the distribution of cheeses according to their physico-chemical composition (data shown in red). Variations in taste and texture characteristics are shown as variables and projected as supplementary in blue in this figure. Rheological measurements are also included to complete the texture description. Briefly, MD means the inverse of elasticity, Df deformability, Cf and Wf cohesion. Cheeses with the highest fat content (G2) were more soluble, sticky, and smooth but less cohesive (Wf), firm, elastic and grainy, and were perceived as more astringent and less salty. The saltiest cheeses (S2) were more cohesive (Cf, Wf), firm, less smooth and sticky, and perceived as saltier but less sweet and bitter. Regarding the highest whey lactose content (T2), the cheeses had a smoother and stickier texture, but were less cohesive (Cf, Wf), deformable (Df), and elastic (MD and perception), and had more intense tastes except for sweet and bitter. The addition of different strains was linked to differences in salty and sweet tastes and in texture (MD, Df, firm, elastic, sticky).

### 2.3. Microstructure and Image Analysis

The image analysis was performed on cheese produced with one strain, as we assumed that the strain had no effect on microstructure. Several microstructural parameters were extracted from image analyses of cheese samples obtained by confocal laser scanning microscopy, including the average size, area of particles, circularity, and Feret diameter (Table 1). A statistical analysis (Wilcoxon test with a Bonferroni correction) was conducted to determine the influence of fat, the technological process, and salt.

An overall univariate analysis showed a significant effect of each parameter composition (fat (G), whey lactose (T) and salt (S)) on most of the microstructure parameters. For the higher-fat content cheese, the average size, area of particles, and Feret diameter (distance between two tangents on opposite sides of the particle) were significantly greater than those for the low-fat content cheese [29], while circularity was higher for the low-fat content cheese. The Feret diameter, indicative of the shape of the fat droplet, is a complementary parameter of circularity that varies both with the elongation and the roughness of the fat droplet. This means that the fat droplet is less circular, more elongated, and less geometric for the higher-fat content cheese.

Regarding the lactose rate (T), all these microstructural parameters were significantly lower for the lowest lactose content compared to the higher values. For salt content, a higher salt level in the cheeses led to a greater number of fat particles, a higher total and particle area, and increased circularity of the fat particles. The variation in these two parameters did not generate a significant variation in the Feret diameter. This means that these parameters seemed to affect the roughness of the droplet more than its geometry.

### 2.4. Sensory Evaluation of Cheese Aroma

The sensory profile of the 16 cheeses was determined using the 30 aroma descriptors evaluated by the panel (Table 2).

From the QDA analyses, 16 aroma descriptors were found to differentiate the 4 studied factors after the ANOVA (*p* value < 0.05). The most discriminant factors were whey lactose content and fat level (9 and 7 descriptors affected, respectively), followed by strain and salt (5 and 4 descriptors). Lactose level discriminated sweet (cooked milk, caramel, vanilla, mushroom) and sweaty (acidified and sour milk, animal, rancid) notes, whereas a high-fat level showed acidified milk, fruity, caramel, vanilla and yeast vs. oxidized notes, and a low-salt level showed vegetal, mushroom and burnt vs. vanilla notes. Strains opposed green and fermented from rancid notes. Figure 2 illustrates these results.

### 2.5. Identification of Aroma Compounds in Cheese by HS-SPME–GC–MS

The GC–MS analyses of cheese extracts by HS-SPME enabled the identification of 36 aroma compounds (Table 3). These compounds belonged to different chemical classes, including volatile fatty acids VFAs (8), aldehydes (10), alcohols (4), ketones (7), esters (2), sulfurs (4), and hydrocarbon (1), which have been previously described as the principal chemical classes of cheese aroma compounds [30,31].

The distribution of these compounds among the 16 cheeses, according to the 4 factors, i.e., fat (G), whey lactose (T), salt content (S) and nature of strains (A) is depicted in the PCA plot in Figure 2a. The aroma compounds that significantly differentiated the cheeses (ANOVA, results not shown) were included as supplementary variables of sensory descriptors.

The production of flavor compounds was favored under the following conditions: a high content of whey lactose for sulfur compounds and aldehydes, except 3-methylbutanal. Low-fat content mainly promoted the formation of aldehydes. The strain A2 was associated with higher levels of 9 aroma compounds, in particular, 2 VFAs, 2 branched alcohols, heptan-2-one, and 3 sulfur compounds. As expected, the salt had less effect on aroma formation, although more branched VFAs, 2-methylbutanol and hexanal were related to a high-salt level, in contrast to 3-methylbutanal.

Correspondence between small code and large code for cheesesSmall codeC-01C-02C-03C-04C-05C-06C-07C-08C-09C-10C-11C-12C-13C-14C-15C-16Large codeG1T1S1A1G1T1S1A2G1T2S1A1G1T2S1A2G1T1S2A1G1T1S2A2G1T2S2A1G1T2S2A2G2T1S1A1G2T1S1A2G2T2S2A1G2T2S2A2G2T2S1A1G2T2S1A2G2T1S2A1G2T1S2A2G: fat; T: lactose in whey; S: salt; 1: lower level; 2: higher level; A1 and A2: both adjunct strains.

### 2.6. Dynamic Aroma Release from Cheese

The dynamic aroma release was monitored simultaneously using temporal sensory analysis (TCATA, Temporal Check All That Apply) and physicochemical temporal analysis (PTR-ToF-MS, Proton Transfer Mass-Time of Flight-Mass Spectrometry).

As expected [32], a large interindividual variability was observed for the release of aroma compounds in the mouth (PTR-ToF-MS) and for temporal perception (TCATA).

Furthermore, fat, salt and lactose levels had a significant effect on the release of most aroma compounds, according to the Wilcoxon test. The type of flavoring strain had an impact on the release of only a few aroma compounds.

#### 2.6.1. TCATA

The TCATA analyses revealed that milky and salty descriptors were primarily present at the beginning of consumption.

The ANOVA carried out on citation durations in TCATA revealed that 3 attributes were significantly more frequently cited: milky (F = 3.46, *p* < 0.0001), salty (F = 3.86, *p* < 0.0001), and bitter (F = 4.17, *p* < 0.0001). The other attributes were cited throughout the sensory evaluation with varying percentages of citations, depending on the cheese, but were not significant.

Figure 3 displays the attribute duration for the TCATA analyses. The first axis of the PCA separated the fat levels. A low-fat level seemed to be correlated with a longer duration of bitter, milky, and salty perceptions.

In this manuscript, we will solely focus the discussion on aroma perceptions while salty perception will be the subject of another manuscript.

#### 2.6.2. PTR-ToF-MS Analyses

Regarding the aroma release with PTR-ToF-MS analyses, 23 ions corresponding to volatile compounds identified with GC–MS or their fragments could be followed during cheese consumption. The ions and their characteristics are detailed in Table 4.

The first axis of the PCA (Figure 4) seemed to separate the fat levels (G1 and G2): a high-fat content seemed to be correlated to a greater release of most flavor compounds. The G1T1S1A1 and G1T1S1A2 products seemed to differ from the other cheeses by the expression of ions 81.069 and 137.112, associated with terpenes, as well as ions 63.027 (dimethyldisulfide), 63.050 (unknown), and 65.023 (fragment) (Figure 4). Furthermore, fat, salt, and lactose levels exhibited significant effects on the release of most aroma compounds. However, the nature of the flavoring strain only influenced the release of a few aromas. This is the reason why we pooled the A1 and A2 data for the same cheese in the following analyses.

### 2.7. Multivariate Analyses

Several multivariate analyses were conducted on the image analyses, PTR-MS data, and TCATA data. Figure 5 illustrates the results of the multiple factorial analysis (MFA).

Axis 1 on the MFA separated the cheese on fat content, while axis 2 is more complex. Considering each cheese individually, G2T1S1 is likely to be different from the others due to a Feret diameter and average size of fat particles being larger than those of the other cheeses; G2T1S2 exhibits a larger surface area of fat particles; G1T1S1 and G1T1S2 are characterized by a higher circularity of these particles. In contrast, G1T2S1 is characterized by a smaller particle size.

Regarding the temporal release of volatile compounds, two cheeses clearly stand out from the others: G1T1S1 and G2T1S1. G1T1S1 has high positive scores on the first and second axes. According to the correlation circle, the first and the second axis are highly correlated (positively) with salty, milky, and spicy but also with a group of ions (high scores for *m*/*z* 49.01, 51.00, 41.03, 67.05, 70.07, 45.03, 47.04, 65.06, 63.02, 63.03 and even higher scores 63.05, 137.1, 81.06, 73.06, and 93.07). Consequently, G1T1S1 is likely to have high scores on these variables. Similar reasoning can be applied to the other products, showing that G2T1S2 is likely to have high scores on ions *m*/*z* 43.05, 89.05, 43.06, 71.08, 117.0, 117.1, 87.04, 131.1, and on the duration of roasted perception. G1T2S1 is likely to have high scores on *m*/*z* 97.06, and on the duration of perception of vegetal, rancid, and fruity attributes. Moreover, G1T2S2 is likely to have a high score on *m*/*z* 61.02 whereas G2T2S2 is likely to have high scores on ions *m*/*z* 87.08 and 87.09, as well as with the duration of roasted perception.

## 3. Discussion

The variation in the fat/dry matter ratio was expected to modify the texture of cheeses and possibly the extraction of aroma compounds in the mouth. Indeed, it induced an expected opposition between firm/cohesive and smooth. Additionally, the lower fat content was linked to higher amounts of some aroma compounds, mainly because fat may impair the extraction of hydrophobic aroma compounds. Interestingly, these compounds were predominantly aldehydes, reflecting an oxidized state of degradation of compounds of fat (linear aldehydes) or amino acids (ramified aldehydes and benzaldehyde). This matches with the more intense oxidized descriptor noted in the corresponding cheeses.

The variations in cheese pH were also expected to modify the texture through proteolysis. However, a lower pH level led to cheeses that were less cohesive, without any apparent correlation with the level of proteolysis. These cheeses were perceived as more acid, pungent and acidified milk, which was expected. Their more intense notes of rancid, sour and sweat, and of animal, sulfur, and vegetable matched with their higher contents of aldehydes and of sulfur compounds, respectively.

The variation in salt content was expected to modify both the texture and the perception of saltiness that could potentially interact with aroma perception. Hence, the higher cohesiveness of the saltier cheeses is in accordance with the findings of Lawrence et al. [35] and the higher salty perception was expected. The higher levels of aldehydes and sulfur compounds of these cheeses may be attributed to a salting-out effect, which enhances the release of these compounds from the matrix.

Furthermore, the variations in strains of secondary microflora were expected to modify the composition of aroma compounds due to differences in metabolic functionalities. In these cheeses, the strain A2 showed a more intense catabolism of amino acids, as suggested by the repartition of branched alcohols and sulfur compounds.

Numerous studies have demonstrated that amino acid degradation is a key process in aroma formation in cheese. The ability of lactic acid bacteria and other cheese microorganisms to degrade amino acids into aroma compounds is highly strain-dependent. Thus, aromatic amino acids (such as phenylalanine, tyrosine, and tryptophan), branched-chain amino acids (including leucine, isoleucine, and valine), and methionine serve as major precursors for certain aroma compounds in cheese [36].

Image analysis of fat droplets in the cheeses provides a potential explanation for the rheological behavior observed in the different cheeses. Larger fat droplets result in a less organized network, leading to a less resistant food matrix. Moreover, the circularity, and consequently the Feret diameter, are significantly affected by the fat ratio. It is notably higher for cheeses with lower-fat content values.

Regarding whey lactose content, lower lactose levels correspond to a less firm texture due to greater proteolysis. This observation is confirmed by rheological data and impacts all the microstructural parameters. When lactose levels increase, the cheese matrix becomes significantly firmer. Surprisingly, although this treatment mainly affects proteolysis, it has a significant impact on all fat microstructural parameters. The decrease in all these parameters when proteolysis is lower and firmness is higher, indicates that when the protein network is denser, the fat particles become more confined in the network, undergo more stress, tend to reduce their size and adopt a form according to the space available for them in the network.

Concerning salt content, only the percentage area occupied by fat was affected, with a higher value observed for the highest content in salt. This suggests a greater number of particles in this case, but with the same shape regardless of the salt content.

The composition of the food matrix, along with its rheological properties and microstructure, is known to influence the release of volatile compounds in the mouth during eating and this influence varies according to the physicochemical properties of the volatile compounds [37]. In this study, a main objective was to try to correlate composition, microstructure, release of volatile compounds and temporal perception when eating real cheeses. To date, most studies have focused on model products or products with uncontrolled variations in composition. The main interest of our study is that it concerns real cheeses with controlled compositions and structures.

Concerning the release of volatile compounds overall, we observed that cheeses with a low-fat content release more hydrophobic volatile compounds compared to cheeses with a higher-fat content. This seems logical in that the release of hydrophobic compounds is lower in the vapor phase from oil than release from a water phase [38].

This observation is rather consistent with the results obtained by Tarrega et al. [39] who reported a significant decrease in release parameters for most of the aroma compounds when increasing the lipid/protein ration in a model cheese. They also highlighted substantial interindividual variability in both flavor release and chewing behavior. However, these results seem to be in opposition to those reported by Boisard et al. [8] on cheese matrices of the same type. In fact, they observed a significant decrease in release (both quantity and maximum concentration) for the most hydrophobic aroma compounds as the lipid/protein ratio decreased. They attributed their finding to a higher retention of hydrophobic compounds in the protein network, which was thicker and stronger for a lower lipid/protein ratio, and by the microstructure of these model cheeses, which was more rigid and contained more circular fat droplets, contributing to a more stabilized system that limited the diffusivity of these aroma compounds. It can be observed that the difference between the two studies is the amplitude of the lipid/protein ratio, which is larger (1 and 0.5) in the Tarrega study. This difference can lead to different microstructures and molecular organizations of fat droplets and protein networks, which strongly impact the availability of aroma compounds during the oral process.

Overall, the circularity of fat droplets is very well correlated with the quantities of small volatile compounds released and rather polar molecules such as ethanol, dimethyl disulfide, acetaldehyde, while it is anticorrelated with the quantities of methyl butanal, piperazine, hexanal and heptanal released, as these are less polar and have a higher molecular mass. This observation seems rather in line with previous results [8] reporting that more rigid cheese models with more circular fat droplets contributed to a more stabilized system that could limit the diffusivity of the more hydrophobic volatile compounds.

The release of a significant number of volatile compounds for G1T1S1, such as acetaldehyde, ethanol, methanethiol, dimethyldisulfide, terpenic compounds, and methylthioacetate, was very well correlated with milky, salty and sharp notes. This indicates that these volatile molecules contribute, likely at different levels, to the complex milky note. The correlation with the salty note could be explained by the cognitive association between milky and salty notes, which enhances the perception of saltiness [40,41]. For G1T2S1, the vegetal note is well-correlated with one unidentified compound present in too low a quantity, which seems to be mainly involved in this note. Finally, the roasted note is well-correlated with certain volatile compounds for the G2T1S2, such as butane-2,3-dione, heptan-2-one, hexanoic acid, 2-methyl-propanoic acid, but with 3-methylbutanal and piperazine for G2T2S2. These volatile compounds do not necessarily have a roasted odor but this note may result from the context of a mixture of several odorous compounds with the aromatic result being reminiscent of a roasted aroma. This remains to be verified in further work.

In this study, it is quite surprising to see that certain correlations between composition, microstructural characteristics, the release of odorous compounds, and temporal perception are valid for a cheese and seem almost specific, since the correlations observed are different from one cheese to another. For some cheeses, we observed no correlation, and we cannot determine which composition or structure factor is responsible. These relationships are more difficult to explain and seem much more complex than those for similar studies carried out on model cheeses. An explanation may be that in model cheeses, the aroma is added during the cheese process just before the solidification of the product, while in our study on real matured cheeses, the odorous compounds are synthesized gradually during maturation during which, at the same time, fat and proteins are profoundly modified in composition and structure, which probably leads to different interactions and distributions of compounds. These factors may explain these phenomena, which will need to be elucidated in subsequent studies.

## 4. Materials and Methods

### 4.1. Fabrication of the Cheeses

Sixteen pressed uncooked raclette-type cheeses were produced from pasteurized milk under controlled conditions in the mini experimental cheese-making plant at PAM INRAE (Poligny). Several characteristics of the flavor and structure of the cheeses were controlled by varying 4 factors (2 levels for each composition factor): cheese fat/dry matter content was controlled by partial skimming of milk; whey lactose content was controlled by a water dilution of the whey/curd mix in the vat; salt/water content was controlled by different times of brining and the use of 2 different strains of inoculated lactic acid bacteria (*Lactococcus lactis* ssp. *lactis* biovar. diacetylactis). The pH of the 16 cheeses was 5.19 ± 0.08. The detailed composition of the cheeses, along with their codes, are presented in Table 5.

The cheeses were ripened for 10 weeks at 12 °C, then stored at 4 °C until the different analyses were conducted in the 11th week.

### 4.2. Description of Aroma, Texture and Tastes of Cheese by Sensory Analysis

For the sensory and TCATA analyses, all panelists were informed and signed a consent form to participate in the study, which was conducted following the Helsinki Declaration. Moreover, an ethical committee approved this study (INSERM (French National Institute for Health and medical Research) Ethic Evaluation Committee N° 20-754, approved in December 2020). Participants with food allergies were excluded from the study.

A quantitative descriptive analysis (QDA) of the 16 different cheeses was realized with 12 trained panelists from the PAM unit over 8 sessions. The composition of the panel was as follows: 8 women and 4 men, aged between 31 to 78. Prior to cheese evaluation, 4 training sessions were conducted to train the judges to recognize odors and tastes with the pure products, to score these in the cheeses, and to build the evaluation sheet. During each evaluation session, 2 cheeses were evaluated, with 2 replicates. Panelists were asked to evaluate the intensity of each descriptor using a linear scale from 0 (absence) to 10 (intense) for texture, taste, and global aroma intensity, and from 0 to 5 for aroma. Texture descriptors were as follows: Firm, Elastic, Smooth, Soluble, Sticky, and Grainy. Taste descriptors were as follows: Salty, Acid, Bitter, Sweet, Pungent, Metallic, and Astringent. The list of the descriptors is presented in Table 6.

### 4.3. Rheological and Microstructure Characteristics of the Cheeses

The rheological characteristics of the cheeses were assessed by a uniaxial compression test at a constant displacement rate with a TX-TA2 texturometer (Stable, Micro Systems Ltd., Champlan, France), following the method described by Lawrence et al. [16]. The four following parameters were recorded: the modulus of deformability (MD, kPa) representing elasticity; the fracture strain Df (dimensionless) representing deformability; the fracture stress (Cf, kPa); and the work to fracture (Wf, kJ·m^−3^) representing cohesiveness.

Moreover, images of cheese microstructures were taken with a Leica TCS SP8 inverted confocal laser-scanning microscope (Leica microsystem, Heidelberg, Germany). The sample was observed using an oil-immersive ×40 lens with a pinhole diameter at 1 Airy Unit. The excitatory wavelength was at 488 for Nile Red (Sigma-Aldrich Chimie Sarl, St Quentin Fallavier, France) and 552 nm for Orange Acridine (Invitrogen, Thermo Fisher Scientific, Illkirch-Graffenstaden, France) and the emission filters were set at 573 nm–743 nm for Nile Red and 494 nm–546 nm for Orange Acridine. The images were treated with the software LasX (version3.5.621594, Leica Microsystemes SAS, Nanterre, France) and Fiji (ImageJ, version 1.52h, NIH, Bethesda, MD, USA). Image analysis was performed on portions of 5 × 5 × 2.5 mm. The dimension of the 2D images was 290.62 × 290.62 µM and each image results from the maximum projection of 9 planes of 0.375 µM thickness each, or 3 µM thickness in total. The resolution of the images was 1024 × 1024 pixels. The mean area, the percentage of area, the circularity, and the Feret Diameter of the fat droplets were evaluated. The circularity was calculated with the formula 4π × area/perimeter2, for which a value of 1.0 indicated a perfect circle. The percentage area occupied by fat in the micrograph corresponded to the percentage of the total area occupied by fat droplets compared to the entire area. The Feret diameter was calculated as the distance between two tangents on opposite sides of the fat droplet.

### 4.4. Identification of Aroma Compounds in Cheese by HS-SPME-GC-MS

Both neutral VOCs and volatile fatty acids (VFAs) were analyzed using solid-phase microextraction–gas chromatography–mass spectrometry (SPME–GC–MS). Five grams of grated cheese were ground with an ultra-turrax (4 × 40 s) in 45 g UHQ water.

Neutral VOCs: 3 mL of the cheese suspension were transferred to a 10 mL vial. The sealed vial was equilibrated at 40 °C for 30 min. The SPME fiber (85 μm carboxen/polydimethylsiloxane (CAR/PDMS); Supelco, Saint Quentin Fallavier, France) was exposed to the headspace for 40 min. Then, it was inserted into the splitless/split injector at 250 °C (5 min in splitless mode) of a gas chromatograph (HP6890 Agilent Technologies, Les Ulis, France) equipped with a fused-silica capillary RXI-5MS column (60 m, 0.32 mm i.d., film thickness 1 μm; Restek, Lisses, France). The carrier gas (helium) flow was set at 2 mL·min^−1^. The GC oven temperature was programmed from 40 °C to 250 °C (6 °C·min^−1^ until 145 °C, 20 °C·min^−1^ until 250 °C holding 1 min). Mass spectrometry was conducted using a mass selective detector (MSD 5973; Agilent Technologies) in electronic impact mode (70 eV). The masses were scanned from *m*/*z* 29 to 206. The ion source temperature was maintained at 230 °C. The identification of VOCs was carried out with the NIST spectra library and by comparing calculated Kovats retention indexes (KIs) with those of standard compounds and data in the literature (PubChem, https://pubchem.ncbi.nlm.nih.gov/docs/compounds, accessed on 30 March 2024). The n-alkanes used to calculate KI were found directly in the cheese chromatograms. A semi-quantification was performed by recording the area of the specific ion of each compound in the arbitrary mass unit (amu).

VFAs: 1 mL of cheese suspension was transferred to a 10 mL vial and 300 µL of H_2_SO_4_2N was added. The sealed vial was equilibrated at 60 °C for 12 min. The SPME fiber (75 μm carboxen/polydimethylsiloxane (CAR/PDMS) fiber; Supelco Saint Quentin Fallavier, France) was exposed to the headspace for 20 min. Then, it was inserted for 5 min into the splitless/split injector at 240 °C in the above-cited GC–MS apparatus. The fused-silica capillary DB-Wax column (30 m, 0.32 mm i.d., film thickness 0.5 μm; Agilent, Les Ulis, France) was flushed with helium at 2 mL·min^−1^. The GC oven temperature was programmed from 120 °C to 230 °C (6 °C·min^−1^ until 180 °C, 10 °C·min^−1^ until 230 °C, holding for 2 min). The MS conditions were as described above except that KIs were not used.

### 4.5. Dynamic Aroma Release Study

The judges in this study were different than those participating in the QDA analyses. The panel of 12 judges was composed of 10 women and 2 men aged between 20 and 60 years old. Prior to each session, the judges were asked not to drink coffee, eat, or smoke at least 1 h before the session, so as to not alter sensory perception and disturb the PTR-MS acquisition (aromas already present in the breath of the judge). The judges took part in the study, after having been informed and having signed a consent form. During each session, judges evaluated 2 cheeses with 3 replicates of each product, a protocol necessitated by the cheese-ripening process. For the dynamic aroma release assessment, cheeses were cut in one piece, weighing 7.0 ± 0.2 g.

The session of dynamic aroma release consisted of a nosespace analysis, an analytical chemistry analysis using a PTR-ToF-MS instrument coupled with a temporal sensory analysis (TCATA).

#### 4.5.1. TCATA

The selection of various attributes was based on the key descriptors of the QDA analysis conducted earlier. At the beginning of the study, the judges were familiarized with the odors and tastes with reference solutions (Table 7) for the different attributes of TCATA analysis (Table 8). The attributes for TCATA analyses were selected from the QDA analyses.

For each judge, the attributes were put in the same order during the whole sensory evaluation. However, their orders were randomized among the panelists to prevent them from preferentially choosing attributes from the top of the list [32].

The TCATA analyses (acquisition, data processing) were performed using TimeSens 1.0 (INRAE, Dijon, France) software.

#### 4.5.2. PTR-ToF-MS Analyses

All the nosespace analyses were performed using a PTR-ToF-MS instrument (Ionicon 8000, Ionicon Analytik GmbH, Innsbruck, Austria) upgraded with the ion funnel to better focalize the ions into the ToF mass analyzer. The H_3_O^+^ ions were used as reagent ions. Parameters of the PTR-ToF-MS instrument were as follows: drift pressure of 2.3 mBar, drift temperature of 80 °C, and drift voltage of 390 V, resulting in an electric field strength to number density ratio (E/N ratio) of 117 Townsend (Td, 1 Td = 10^−17^ V·cm^2^). Data were collected using the TofDAQ software provided by the manufacturer of the PTR-ToF-MS. Data acquisitions were performed at 1 mass spectrum ranging from *m*/*z* 0 to 227 per 0.100 s. The nosespace sampling was conducted through a home-made teflon nosepiece that connected both nostrils of the subjects via a light helmet to the PTR-MS transfer line maintained at 110 °C. Nosespace sampling was performed at a total flow rate of 400 mL·min^−1^. The helmet allowed subjects to freely move their heads during the experiments.

#### 4.5.3. Protocol of Dynamic Aroma Release Study

TCATA and nosespace analyses were done simultaneously and required individual sessions that were conducted in an air-conditioned room at 22 °C (±1). Each session lasted around 45 minutes. During the evaluation session, subjects were connected to the PTR-MS instrument. They were asked to evaluate a warm-up sample that preceded the 6 samples (2 cheeses with 3 replicates). The presentation orders were set up following a Williams Latin square experimental design balancing order and position effects.

The subjects were asked to taste the cheese samples according to a free chewing protocol, as naturally as possible to ensure better repeatability of the results.

The protocol of consumption consisted in waiting 30 s before putting the first sample in the mouth, allowing the PTR-MS to record the blank of the composition of the air from the nasal cavity. The TCATA evaluation started after the panelists took the sample in their mouths and clicked on the “Put in the mouth” button displayed on the computer screen. Then, they were required to chew the sample while selecting the perceived attributes as a function of time and select all descriptors that they perceived during the cheese chewing. If they did not perceive a descriptor or several descriptors, they were required to deselect the corresponding descriptors. When the panelists had no more product in their mouth, they were asked to click on the “I have nothing left in my mouth” button. When the panelists no longer perceived any aroma and/or taste, they were asked to click on the “I did not perceive any change” button. The PTR-MS acquisitions were stopped when the signals of the aromas returned to the baseline.

### 4.6. Data Analysis

Data analyses were performed using XLSTAT (statistical and data analysis solution, 2022.3.2, Addinsoft, Paris, France) software. The R (4.0.2) software was also used for TCATA and PTR-MS data, with a specific home R package (PTRMSR) [42] available on github (www.github.com/ChemoSens/PTRMSR, -commit 7d7df23-, accessed on 30 March 2024) and rgcca package [43] (https://github.com/rgcca-factory/RGCCA, R package version 3.0.3, accessed on 30 March 2024).

## 5. Conclusions

The use of real cheese with controlled composition parameters enabled us to characterize how cheese compositions affects aroma compound production, release, and perception. The four basic technological parameters studied (fat content, whey lactose, salt, strains) had some effects on the flavor composition, structure and microstructure, and flavor release of the cheeses studied. The fat content, whey, lactose, and salt have effects on the microstructures of the cheeses, which indirectly impact flavor release. However, the salt content had less influence on the formation of aroma compounds. These compositional factors also influence the availability of aroma compounds and the perception of aroma in different model cheeses but with many notable differences, which somewhat calls into question the use of certain models made more quickly and simply in the laboratory for such studies. Indeed, even if the overall compositions are comparable between the two types of products, significant differences in microstructure, mainly due to maturation, exist, and can alone explain significant differences in texture and the availability of stimuli in the mouth over chewing time and perception. In addition, differences in aroma perception may also be related to the ways in which aroma compounds are generated in the cheese matrix. In one case, they are introduced during the entire process, in another case, which is that of real cheeses, they are generated gradually during maturation by the microorganisms inside this matrix. Regarding the two strains, their impact on the aromatic quality of the cheeses differs little, whatever their composition. A relevant choice of strains, in relation to the composition of the product, should make it possible to improve this quality.

Thus, these results could help to formulate real cheeses with reduced fat or salt contents, or by controlling the texture while keeping the same aroma perception. Better knowledge of the influence of composition factors on the availability of aroma compounds in the mouth and on perception, in connection with the structure and microstructure of real cheeses, should help cheese cheesemakers in the formulations of their products, and also in the development of new quality products well-appreciated by the consumer.

## Figures and Tables

**Figure 1 molecules-29-03412-f001:**
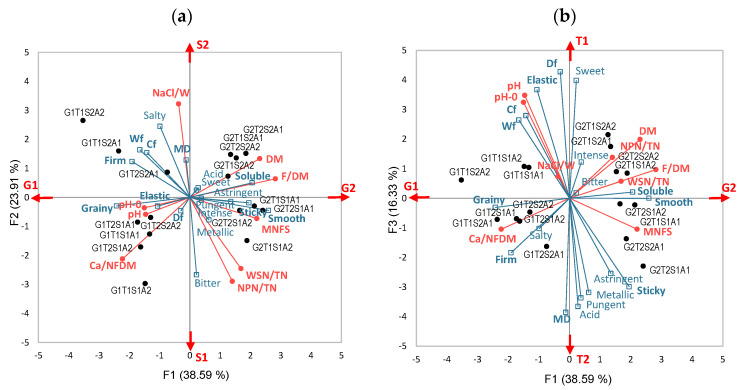
Principal component analysis (PCA) of the physico-chemical composition of cheeses and quantitative descriptive analysis (QDA) data: (**a**) biplot (1–2) of the observations (16 cheeses in black) and the variables active (red: chemical) and supplementary (blue: taste and texture descriptors and rheological measurements); and (**b**) biplot (1–3) of the same observations and variables.

**Figure 2 molecules-29-03412-f002:**
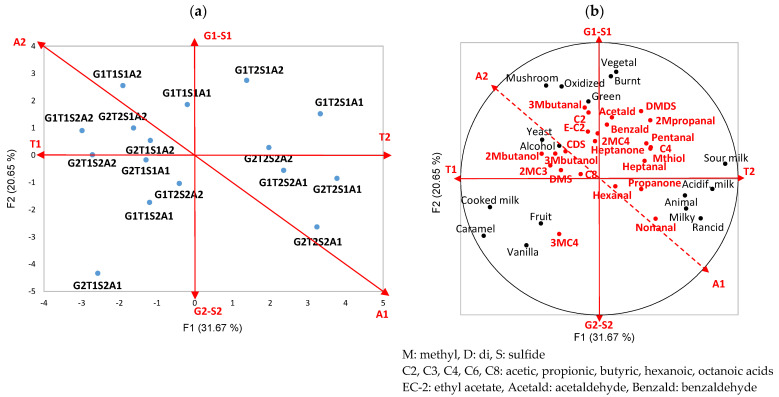
Principal component analysis (PCA) on QDA and GC–MS data: (**a**) PCA individual map (1–2) of the 16 cheeses as observations; and (**b**) PCA correlation circle of the 16 cheeses on sensory descriptors (active variables in black) and aroma compounds identified by GC–MS (supplementary variables in red). G: fat; T: lactose in whey; S: salt; 1: lower level; 2: higher level; A1 and A2: both adjunct strains.

**Figure 3 molecules-29-03412-f003:**
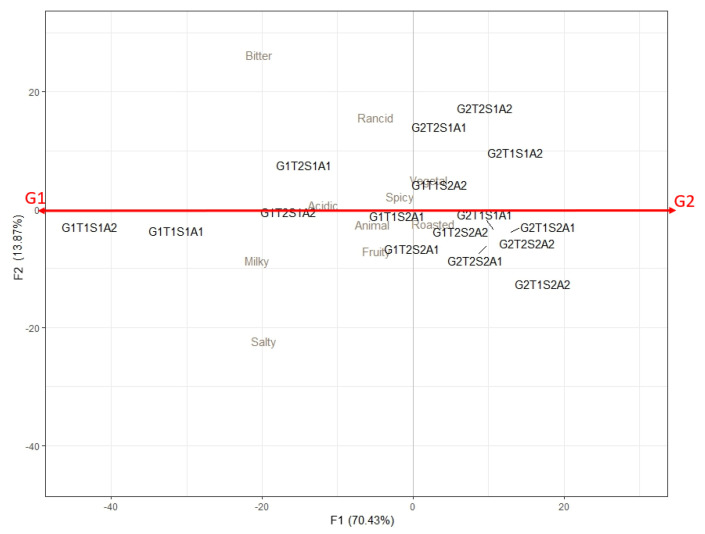
PCA of attribute citation duration. G: fat; T: lactose in whey; S: salt; 1: lower level; 2: higher level; A1 and A2: both adjunct strains.

**Figure 4 molecules-29-03412-f004:**
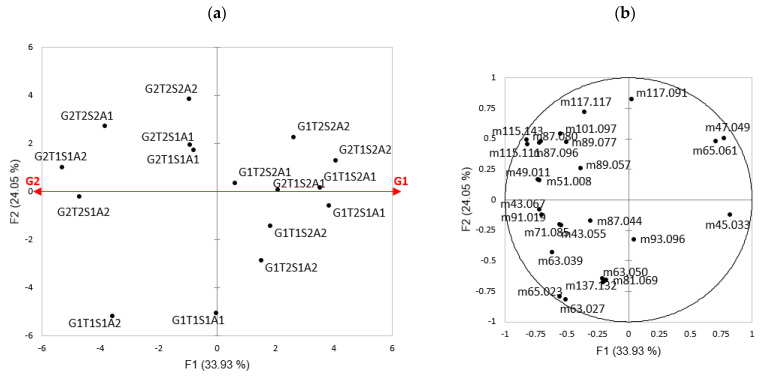
Principal component analysis (PCA) on PTR-MS data: (**a**) individual map (1–2) of the PCA of the 16 cheeses on PTR-MS data; and (**b**) PCA correlation circle of the 16 cheeses on aroma compounds (*m*/*z*). G: fat; T: lactose in whey; S: salt; 1: lower level; 2: higher level; A1 and A2: both adjunct strains.

**Figure 5 molecules-29-03412-f005:**
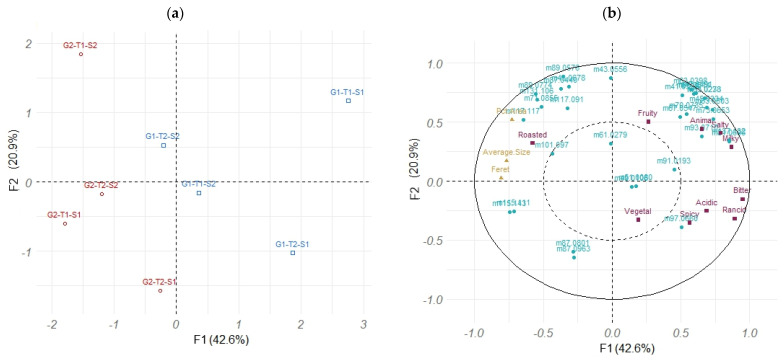
Multiple Factorial analysis (MFA): (**a**) individual map (1–2) of the MFA of the 16 cheeses on PTR-MS data, TCATA data and image data; and (**b**) MFA correlation circle of the 16 cheeses on aroma compounds (*m*/*z*), sensory descriptors, and image analyses. G: fat; T: lactose in whey; S: salt; 1: lower level; 2: higher level.

**Table 1 molecules-29-03412-t001:** Microstructural characterization of the cheeses.

Product	Average Size (µm^2^)	Particle Area (%)	Circularity	Feret Diameter
G1T1S1	16.55	28.29	0.864	5.11
G1T1S2	16.15	29.45	0.867	5.05
G1T2S1	14.4	19.49	0.829	4.99
G1T2S2	17.68	27.62	0.833	5.34
G2T1S1	24.35	32.43	0.805	6.24
G2T1S2	20.77	38.18	0.848	5.73
G2T2S1	17.47	29.20	0.820	5.48
G2T2S2	17.97	32.64	0.828	5.46
Statistical results				
Mean-G1	15.96	25.53	0.848	5.1
Mean-G2	19.89	32.69	0.825	5.7
*p*-value (G)	***	***	***	***
Mean-T1	19.93	32.31	0.843	5.6
Mean-T2	16.89	27.63	0.827	5.33
*p*-value (T)	***	*	***	-
Mean-S1	18.26	27.53	0.828	5.48
Mean-S2	18.34	32.73	0.842	5.44
*p*-value (S)	-	***	*	-

*** < 0.001, * < 0.05, - ≤ 1. G: fat; T: lactose in whey; S: salt; 1: lower level; 2: higher level.

**Table 2 molecules-29-03412-t002:** Main sensory attributes of the cheeses.

Cheese Code	Main Sensory Attributes
G1T1S1A1	Nut
G1T1S1A2	Burnt
G1T2S1A1	Roasted
G1T2S1A2	Milky
G1T1S2A1	Cooked milk, citrus fruit
G1T1S2A2	Oxidized
G1T2S2A1	Sulfur, animal
G1T2S2A2	Alcohol, citrus fruit
G2T1S1A1	Caramel, mushroom
G2T1S1A2	Cooked milk, fresh milk, vanilla
G2T2S2A1	Toasted, roasted, sulfur, rancid
G2T2S2A2	Yeast, alcohol
G2T2S1A1	Acidified milk, sour milk, nut, sweat
G2T2S1A2	Citrus fruit, green vegetal, yeast, alcohol
G2T1S2A1	Cooked milk, fresh milk, vanilla
G2T1S2A2	Fruity, caramel, vegetable, sulfur

G: fat; T: lactose in whey; S: salt; 1: lower level; 2: higher level; A1 and A2: both adjunct strains.

**Table 3 molecules-29-03412-t003:** List and semi-quantification (amu) of aroma compounds identified by HS-SPME–GC–MS.

No.	Volatile Compounds	CAS	KI_exp_	KI_lit_	C-01	C-02	C-03	C-04	C-05	C-06	C-07	C-08	C-09	C-10	C-11	C-12	C-13	C-14	C-15	C-16
	Acids																			
1	Acetic acid	64-19-7	-		6.26	6.62	6.15	6.49	6.38	6.51	6.39	6.54	6.08	6.34	6.36	6.54	6.23	6.45	5.94	6.43
2	Propanoic acid	79-09-4	-		3.37	3.63	3.18	3.44	3.91	3.59	3.92	3.52	3.22	3.53	3.71	3.54	3.71	3.50	2.49	3.07
3	Butanoic acid	107-92-6	-		5.88	5.94	5.94	5.94	5.89	5.92	5.94	5.92	5.98	6.07	5.94	5.98	6.04	6.28	5.99	6.00
4	Hexanoic acid	142-62-1	-		5.52	5.61	5.47	5.56	5.58	5.67	5.58	5.57	5.58	5.67	5.57	5.54	5.60	6.03	5.58	5.57
5	Octanoic acid	124-07-2	-		5.00	5.13	5.05	5.16	5.03	5.20	5.15	5.19	5.00	5.08	5.09	5.05	5.07	5.27	4.98	5.00
6	2-Methylpropanoic acid	79-31-2	-		3.22	3.70	1.82	3.24	4.05	4.11	3.98	3.81	3.67	3.73	4.16	3.91	3.90	3.70	3.61	3.98
7	2-Methylbutanoic acid	116-53-0	-		3.87	4.26	3.27	3.87	4.46	4.60	4.25	4.29	4.13	4.19	4.52	4.25	4.24	4.37	4.18	4.48
8	3-Methylbutanoic acid	503-74-2	-		5.66	5.86	5.26	5.55	6.08	5.93	5.94	5.79	5.71	5.65	6.02	5.72	5.74	5.61	5.57	5.73
	Aldehydes																			
9	Acetaldehyde	75-07-0	<500	391	5.50	5.44	5.36	5.30	5.36	5.35	5.52	5.52	5.27	5.26	5.09	5.55	5.28	5.31	5.15	5.26
10	Butanal	123-72-8	602	585	3.93	5.44	5.06	5.17	4.77	6.16	6.48	5.24	4.61	4.59	5.26	5.91	5.12	5.69	3.81	3.30
11	Pentanal	110-62-3	698	698	4.44	5.41	5.07	5.11	4.88	4.85	5.04	6.43	4.88	5.06	5.12	4.99	4.88	5.05	3.50	3.15
12	Hexanal	66-25-1	800	797	4.40	5.73	4.91	5.53	5.74	5.71	6.68	6.65	5.01	5.27	5.29	5.69	5.33	5.60	4.90	4.80
13	Heptanal	111-71-7	901	897	3.86	5.07	4.35	4.48	4.35	4.40	5.18	6.03	3.67	4.35	4.61	4.59	4.49	4.55	3.47	3.26
14	Nonanal	124-19-6	1100	1103	3.71	4.42	4.43	4.47	4.29	4.16	4.58	4.80	4.22	4.40	4.71	4.44	4.61	4.56	4.35	4.32
15	2-Methylpropanal	78-84-2	546	552	4.52	5.11	4.55	4.70	5.04	4.71	4.76	4.50	3.98	3.55	4.34	5.05	4.57	4.73	3.12	3.15
16	3-Methylbutanal	590-86-3	653	652	5.77	6.33	5.12	5.46	5.33	5.56	5.22	5.28	5.64	5.37	5.35	5.21	5.43	5.18	4.95	4.90
17	2-Methylbutanal	96-17-3	663	660	4.61	4.66	4.61	4.61	4.83	4.61	4.61	4.61	4.89	4.25	4.76	4.70	4.96	4.25	4.24	3.09
18	Benzaldehyde	100-52-7	996	964	4.09	5.57	4.61	4.17	5.49	4.40	4.82	4.48	3.26	3.59	3.39	5.13	3.44	4.68	3.06	3.00
	Alcohols																			
19	Ethanol	64-17-5	<500	412	6.85	7.28	7.02	7.07	7.01	6.94	7.26	7.28	7.06	6.90	7.04	6.90	6.94	6.98	6.94	6.98
20	Butanol	71-36-3	672	657	5.12	6.40	5.07	5.24	5.71	6.37	5.12	5.12	5.19	4.77	5.61	5.50	5.27	5.31	4.76	4.38
21	3-Methylbutan-1-ol	123-51-3	737	738	5.89	7.34	5.77	6.76	6.68	7.39	6.66	7.45	6.70	7.25	6.62	7.30	6.53	7.39	6.11	6.53
22	2-Methylbutan-1-ol	137-32-6	739	740	5.10	6.89	4.46	5.87	5.90	6.89	5.69	6.62	5.62	6.05	5.66	6.16	5.56	6.25	5.24	5.46
	Ketones																			
23	Propan-2-one	67-64-1	<500	479	5.15	5.53	5.70	5.56	5.42	5.61	6.05	5.94	5.58	5.52	5.53	5.77	5.66	5.87	5.51	5.44
24	Butan-2-one	78-93-3	610	587	5.17	6.37	5.14	5.63	5.82	5.41	5.57	4.88	4.64	4.75	4.90	5.87	4.90	5.44	3.47	4.50
25	Pentan-2-one	107-87-9	700	679	3.72	5.09	6.04	5.51	4.68	4.14	5.13	5.22	5.77	5.82	5.07	5.17	5.46	5.99	4.79	4.72
26	Heptan-2-one	110-43-0	888	888	4.66	5.95	5.47	5.71	5.44	5.55	5.62	5.77	5.51	5.74	5.36	5.49	5.42	5.66	5.12	5.34
27	Nonan-2-one	821-55-6	1091	1085	3.75	5.29	4.52	4.84	4.67	4.79	4.85	5.10	4.68	4.76	4.30	4.41	4.29	4.42	3.97	4.44
28	Butane-2,3-dione	431-03-8	602	596	6.00	6.58	4.07	5.38	5.74	5.17	5.02	4.66	5.93	5.77	6.17	4.98	4.64	4.95	4.44	4.99
29	Acetoin	513-86-0	719	713	6.16	6.63	6.53	6.63	6.97	6.67	7.07	6.44	6.58	6.85	6.86	5.49	6.51	5.66	5.81	6.00
	Esters																			
30	Ethyl acetate	141-78-6	615	609	4.66	6.28	4.11	5.18	5.69	5.41	5.33	5.37	4.94	4.61	4.95	4.99	4.57	4.98	3.69	4.08
31	Ethyl butanoate	105-54-4	799	798	3.87	5.17	5.01	5.32	5.18	5.21	6.02	5.80	5.21	5.13	5.07	5.17	5.05	5.21	4.97	4.90
	Sulfurs																			
32	Methanethiol	74-93-1	<500	422	3.76	3.97	4.08	4.13	3.97	4.00	4.73	4.60	4.05	4.21	4.28	4.44	3.93	4.49	3.26	3.94
33	Carbon disulfide	75-15-0	529	549	5.75	5.75	5.48	5.41	5.43	5.37	5.19	5.18	5.28	5.20	5.41	5.66	5.20	5.38	5.62	5.60
34	Dimethylsulfide	75-18-3	511	521	4.46	5.04	4.81	4.92	4.54	4.91	4.71	5.08	5.13	5.43	4.72	5.27	5.15	5.57	5.08	5.25
35	Dimethyldisulfide	624-92-0	747	747	4.57	4.57	4.63	4.63	4.63	4.57	4.63	4.63	3.72	4.57	4.68	4.87	4.80	5.23	3.11	4.87
	Hydrocarbon																			
36	Benzene	71-43-2	661	657	5.84	5.84	5.59	5.66	4.99	5.02	4.16	4.99	4.18	4.12	5.38	6.06	5.22	5.55	4.10	4.67

KI_exp_: Kovats retention Index determined experimentally; KI_Lit_: Kovats retention Index found in the literature (PubChem: pubchem.ncbi.nlm.nih.gov).

**Table 4 molecules-29-03412-t004:** Aroma compounds followed with PTR-ToF-MS during cheese consumption.

Experimental Mass (*m*/*z*, g·mol^−1^)	Chemical Formula [MH]^+^	Expected Mass (*m*/*z*, g·mol^−1^)	Tentative Identification ^1^ or Identified with HS-SPME–GC–MS ^2^	CAS	Ref.
43.055	C_3_H_7_^+^	43.055	Alkyl fragment ^1^		[33]
45.033	C_2_H_5_O^+^	45.033	Acetaldehyde ^2^	75-07-0	[33,34]
47.049	C_2_H_6_OH_+_	47.049	Ethanol ^2^	64-17-5	[34]
49.010	CH_5_S^+^	49.011	Methanethiol ^2^	74-93-1	[33,34]
63.027	C_2_H_7_S^+^	63.026	Dimethyldisulfide ^2^	624-92-0	[33,34]
63.039	C_2_H_7_O_2_^+^	63.044	Acetaldehyde water fragment ^1^		[33]
65.023	C_5_H_5_^+^	65.038	Fragment ^1^		[33]
65.061	C_2_H_9_O_2_^+^	65.060	Ethanol-water cluster ^1^		[33]
71.085	C_5_H_11_^+^	71.086	Fragment (terpene, ester) ^1^		[33]
81.069	C_6_H_9_^+^	81.070	Terpene fragment ^1^		[33]
87.044	C_4_H_7_O_2_^+^	87.044	Butane-2,3-dione ^2^	431-03-8	[33]
87.080	C_5_H_11_O^+^	87.080	3-Methylbutanal ^2^ 2-Methylbutanal ^2^ Pentanal ^2^	590-86-3 96-17-3 110-62-3	[33]
87.096	C_4_H_11_N_2_^+^	87.092	Piperazine ^1^		
89.057	C_4_H_9_O_2_^+^	89.060	Acetoin ^2^ 2-Methylpropanoic acid ^2^ Butanoic acid ^2^ Ethyl acetate ^2^	513-86-0 79-31-2 107-92-6 141-78-6	[33]
89.077	C_5_H_13_O^+^	89.096	3-Methylbutan-1-ol ^2^ 2-Methylbutan-1-ol ^2^	123-51-3 137-32-6	[33]
91.019	C_3_H_7_OS^+^	91.021	Methyl thioacetate ^1^		
93.071	C_7_H_9_^+^	93.069	Toluene ^1^ Terpene fragment ^1^	108-88-3	[33]
101.097	C_6_H_13_O^+^	101.096	Hexanal ^2^	66-25-1	[33]
115.111	C_7_H_15_O^+^	115.112	Heptanal ^2^ Heptan-2-one ^2^	111-71-7 110-43-0	[33] [33]
115.143	C_8_H_19_^+^	115.148	Octane ^1^		
117.091	C_6_H_13_O^+^	117.091	Hexanoic acid ^2^	142-62-1	[33]
117.117	C_7_H_17_O^+^	117.127	Heptan-2-ol ^1^ Heptan-1-ol ^1^	543-49-7 111-70-6	[33]
137.132	C_10_H_17_^+^	137.132	beta-Myrcene ^1^ Limonene ^1^ beta-Ocimene ^1^ 3-Carene ^1^ alpha-pinene ^1^	123-35-3 138-86-3 13877-91-3 13466-78-9 80-56-8	[33]

^1^: Remaining tentative identification of the aroma compounds relies on available PTR-ToF-MS literature; ^2^: Identification of the aroma compounds by comparisons with our HS-SPME–GC–MS analyses; Eleven of the 23 ions (Table 4) were identified and quantified by HS-SPME–GC–MS; for the remaining ions, tentative identification was carried out using references in the literature and with the TofDAQ software (1.2.99).

**Table 5 molecules-29-03412-t005:** Composition of cheeses.

Cheese Code	Fat (G, %)	Whey Lactose (T, g·L^−1^)	Salt (S, %)	Lactic Acid Bacteria (A)
G1T1S1A1	40	33	2.5	A1
G1T1S1A2	40	33	2.5	A2
G1T2S1A1	40	42	2.5	A1
G1T2S1A2	40	42	2.5	A2
G1T1S2A1	40	33	4	A1
G1T1S2A2	40	33	4	A2
G1T2S2A1	40	42	4	A1
G1T2S2A2	40	42	4	A2
G2T1S1A1	50	33	2.5	A1
G2T1S1A2	50	33	2.5	A2
G2T2S1A1	50	42	2.5	A1
G2T2S1A2	50	42	2.5	A2
G2T1S2A1	50	33	4	A1
G2T1S2A2	50	33	4	A2
G2T2S2A1	50	42	4	A1
G2T2S2A2	50	42	4	A2

**Table 6 molecules-29-03412-t006:** List of descriptors used for the quantitative descriptive analysis (QDA).

List of Descriptors		
Fresh lactic	Cooked milk	Milky
Acidified milk	Sour milk	Animal
Fruity	Citrus fruit	Jam
Caramel	Toasted	Burnt
Mushroom	Vegetables	Green
Vegetal	Broth	Yeast
Nut	Soap	Tyre
Oxidized	Alcohol	Chemical
Sulfur	Rancid	Rust
Sweat	Mild roasted	Strong Roasted

**Table 7 molecules-29-03412-t007:** TCATA attributes and sensory references used for the TCATA analysis.

TCATA Attributes	List of Different Aromas Regrouped in the TCATA Attributes	Sensory References
Acid		Lactic acid solution at 1.5 mg·L^−1^ in Evian water
Bitter		Caffeine solution at 0.5 mg·L^−1^ in Evian water
Milky	Milk, cream, yogurt, cottage cheese, boiled milk, melted butter	Cottage cheese
Fruity		Apricot jam
Mild roasted	Hazelnut, vanilla, caramel	Crumbled French biscuit named “Petit Beurre”
Vegetable	Vegetable broth, cut grass, leek, mushroom, earthy	Green vegetable soup
Animal	Leather, horse, sweat	Leather pieces macerated in water
Rancid, soap, sour		Marseille soap shavings

**Table 8 molecules-29-03412-t008:** List of attributes for TCATA analysis.

Taste Attributes	Aroma Attributes
Salty	Animal
Bitter	Vegetable
Acid	Milky
	Mild roasted
	Spicy
	Rancid, soap, sour
	Fruity

## Data Availability

The data presented in this study are available on request.

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
