# Peer review of "Influence of Cheese Composition on Aroma Content, Release, and Perception"

_molecules, 2024, doi:10.3390/molecules29143412_

Round 1

Reviewer 1 Report

Comments and Suggestions for Authors

The study is very interesting and has been carried out to a high standard. The use of nosesence PTR-TOF MS is very novel and working on real cheese instead of models is also very novel and more relevant. I made a few minor queries in the text.

Comments on the Quality of English Language

No major issues

Author Response

The authors thank very much the reviewer for taking the time to review this article, for their constructive remarks and suggestions to improve the manuscript.

I've made the corrections and added the explanations as requested. These additions are highlighted in yellow in the text, and I have summarised the pages and lines of the changes made.

Page 2 line 68: some explanations are added

Page 2 line 71: the word “the” is added

Page 2 line 84: the word “thin” is added

Page 3 line 130: some explanations are added about feret diameter

Page 3 line 188: The volatile fatty acids were replaced by VFAs

Page 14 line 305: The word “cheese” was deleted

Page 14 line 305: The word “cheese” was deleted

Page 14 line 306: The word as is added

Page 17: please clarify why 60C and also which VFA were done by this method as opposed to the method described above?

The VFA method is designed to increase the sensitivity of the analysis. Addition of acid for VFA extraction allows to make all the VFAs in the COOH form, which increases the extraction from fat.

60°C increases the volatility of compounds compared to 40°C. 60°C is not possible for the mix of neutral compounds because of the risk to produce neo-formed compounds from the different chemical classes. It is not a problem with VFAs, in large excess.

Reviewer 2 Report

Comments and Suggestions for Authors

The manuscript entitled "Influence of the cheese composition on the aroma content, release and perception" try to explain the influence of different parameters on the aroma formation. The topic is very interesting and may be helpful in cheese production. However, I have several questions and suggestions.

1. The importance of the study is not highlighted. Can the results be useful for cheese production? Why these cheese parameters should be studied?

2. There are a lot of information on the PCA figures. Is it possible to do CA? Maybe CA will be also interesting and the figures more clear and easier to interpretation?

3. The conlusions are not clear for me. What parameters are the best to obtain optimal cheese product? How the results can be used?

4. The references cited in this manuscript are appropiate and relevant. However, only 8 from 35 cited papers are from the lat 5 years. So 27 of them are quite old and not up-to-date.

Author Response

The authors thank very much the reviewer for taking the time to review this article, for their constructive remarks and suggestions to improve the manuscript.

  1. 1. The importance of the study is not highlighted. Can the results be useful for cheese production? Why these cheese parameters should be studied?

The objective of the study is to determine how flavour perception can be impacted by the composition and structure of the cheese matrix. The fabrication of real cheeses with controlled composition is a challenging way to perform such a study.

The factors we choose to make the composition / structure vary are technological factors that cheesemakers can commonly make vary in their fabrication scheme of semi-hard cheeses: Fat matter content, by the way of more or less skimming the milk; Removal of lactose in the whey by more or less dilution with water, in order to change the acidification and pH in cheese and modify some enzymatic activities; Salt content, by the way of modification of the brining time; Aroma compounds by the choice of different starters. Hence, knowledge of the influence of these factors on flavour compound release and perception can help cheesemakers to formulate their products, in particular new products.

  1. There are a lot of information on the PCA figures. Is it possible to do CA? Maybe CA will be also interesting and the figures more clear and easier to interpretation?

The CA is used for qualitative data. We prefer to use PCAs to remain consistent with the other PCAs for profile intensity, for example (which are quantitative data), and that this allows us to take account of durations rather than quotations. A correspondence analysis will contain just as much information a priori (product, descriptors, ellipses) and will not be any more informative.

  1. The conclusions are not clear for me. What parameters are the best to obtain optimal cheese product? How the results can be used?

As requested by reviewer 3, we have drafted a conclusion. We hope that it is helpful for you.

  1. The references cited in this manuscript are appropriate and relevant. However, only 8 from 35 cited papers are from the last 5 years. So 27 of them are quite old and not up-to-date.

As far as we know, there are no recent studies linking flavour compounds and microstructure in cheeses. The publication founded there are not in link with the subject of our publication.

Reviewer 3 Report

Comments and Suggestions for Authors

Very interesting topic of work. Well-planned selection of research tools; advanced research and sensory analysis. carefully planned receipt of cheese samples, variety of parameters of quantitative and qualitative composition.

Carefully selected keywords

Comprehensively described research team, well introducing the reader to the scope of research and its significance

Fig. 1 values ​​on the axis describing G! should have the same value, especially since it will not reduce readability.

Research methodology described at a high level, allowing experiments to be repeated by other researchers.

Discussion of research, clear and accurate. the discussion confirms the authors' high knowledge of the research topic and substantive discussion with the results of other researchers.

One of the key elements of work submitted to MDPI, including Moleculers, are the conclusions from the work. This was missing from the manuscript under review, and it is a very serious error that absolutely must be corrected.!!!!!

Author Response

The authors thank very much the reviewer for taking the time to review this article, for their constructive remarks and suggestions to improve the manuscript.

Fig. 1 values on the axis describing G! should have the same value, especially since it will not reduce readability.

We did the modification, as requested and had the corrected figures in the text.

One of the key elements of work submitted to MDPI, including Moleculers, are the conclusions from the work. This was missing from the manuscript under review, and it is a very serious error that absolutely must be corrected.!!!!!

We didn't write any conclusions because in the template it says “This section is not mandatory but can be added to the manuscript if the discussion is unusually long or complex.” As requested, we wrote a conclusion in this modified version following the reviewers' comments (please see page 21 lines 570 to 602).

Round 2

Reviewer 2 Report

Comments and Suggestions for Authors

I recommend this paper for publiacation. I have no more questions.